# A Glimpse into the Structural Properties of the Intermediate and Transition State in the Folding of Bromodomain 2 Domain 2 by Φ Value Analysis

**DOI:** 10.3390/ijms22115953

**Published:** 2021-05-31

**Authors:** Leonore Novak, Maria Petrosino, Daniele Santorelli, Roberta Chiaraluce, Valerio Consalvi, Alessandra Pasquo, Carlo Travaglini-Allocatelli

**Affiliations:** 1Dipartimento di Scienze Biochimiche “A. Rossi Fanelli”, Sapienza University of Rome, 00185 Roma, Italy; leonore.novak@uniroma1.it (L.N.); maria.petrosino@uniroma1.it (M.P.); daniele.santorelli@uniroma1.it (D.S.); roberta.chiaraluce@uniroma1.it (R.C.); valerio.consalvi@uniroma1.it (V.C.); 2ENEA CR Frascati, Diagnostics and Metrology Laboratory FSN-TECFIS-DIM, 00044 Roma, Italy; alessandra.pasquo@enea.it

**Keywords:** bromodomain, protein folding, Φ value analysis, protein stability, mutagenesis, folding kinetics

## Abstract

Bromodomains (BRDs) are small protein interaction modules of about 110 amino acids that selectively recognize acetylated lysine in histones and other proteins. These domains have been identified in a variety of multi-domain proteins involved in transcriptional regulation or chromatin remodeling in eukaryotic cells. BRD inhibition is considered an attractive therapeutic approach in epigenetic disorders, particularly in oncology. Here, we present a Φ value analysis to investigate the folding pathway of the second domain of BRD2 (BRD2(2)). Using an extensive mutational analysis based on 25 site-directed mutants, we provide structural information on both the intermediate and late transition state of BRD2(2). The data reveal that the C-terminal region represents part of the initial folding nucleus, while the N-terminal region of the domain consolidates its structure only later in the folding process. Furthermore, only a small number of native-like interactions have been identified, suggesting the presence of a non-compact, partially folded state with scarce native-like characteristics. Taken together, these results indicate that, in BRD2(2), a hierarchical mechanism of protein folding can be described with non-native interactions that play a significant role in folding.

## 1. Introduction

The correct folding process of the biological macromolecules is crucial for living cells, as their biochemical processes rely on finely tuned inter-molecular recognition events, which depend on structural complementarity between interacting molecules. This biochemical principle, known as the structure–function relationship, is particularly evident in the case of structurally complex macromolecules, such as proteins. However, notwithstanding decades of experimental, theoretical, and computational efforts, the mechanism of protein folding is still one of the major problems in molecular biology.

As for any chemical reaction, a clear description of the folding of a protein would require the identification and structural characterization of each of the molecular species transiently populated during the process [1]; however, in the case of protein folding, experimental difficulties arise because of the intrinsic cooperativity of the process and the large number of weak interactions forming from the denatured state to the native state in a biologically relevant timescale [2].

In this context, the description of the folding mechanism of proteins populating partially folded intermediates is particularly valuable, as it may offer the opportunity to follow the evolution of structure formation. In this context, the BET (bromo-extra-terminal domain) bromodomains (BRDs) represent a useful experimental system not only because of their limited molecular weight (about 100 amino acids) and structural simplicity, but also because their role in a variety of patho-physiological processes is becoming increasingly evident [3,4]. Recently, the second BRD of BRD2 (hereafter, BRD2(2)) has been highlighted as an essential node in the cellular response to SARS-CoV-2 infection [5,6].

BRDs are structural motifs that are known to recognize and bind to acetylated Lys residues in histones. The available structures of a variety of BRDs in the PDB database show that this domain consists of a helical bundle composed of four conserved α-helices (αZ, αA, αB, and αC, from the N- to the C-terminus of the domain) connected by loop regions of variable length, notably the ZA loop connecting helices αZ and αA and the BC loop connecting helices αB and αC [7] (Figure 1). These domains act as modulators of eukaryotic gene expression [7,8] by recognizing and binding to the N-terminal tails of histone proteins containing one or more acetyl-lysine residues (AcK). The specific recognition of the post-translationally modified AcK involves a set of highly conserved residues in the hydrophobic core of the domain [9]; however, it has been recently proposed that the highly flexible and less conserved ZA loop may additionally contribute to the binding mechanism [10,11,12].

We recently described the kinetic folding mechanism of two BET bromodomains: the second BRD of BRD2 (BRD2(2)) and the first BRD of BRD4 (BRD4(1)), demonstrating in both cases the population of a transient obligatory intermediate by rapid mixing and temperature jump (un)folding experiments [13]. Here, we take a step further in elucidating the folding scenario of these domains by providing a structural characterization of the intermediate (I) and late transition state (TS2) of the BRD2(2) domain by Φ value analysis [14], as probed by the kinetic analysis of 25 site-directed mutants.

## 2. Results and Discussion

### 2.1. Urea-Induced Equilibrium Unfolding

Twenty-five site-directed mutants were designed and produced in order to perform the Φ value analysis on BRD2(2). The mutants were designed by following the accepted guidelines adopted in Φ value analysis [14,15,16] and characterized by equilibrium denaturation experiments. The thermodynamic stability of the different BRD2(2) mutants was measured by urea-induced equilibrium unfolding experiments [13] at a pH of 7.5 and 20 °C by monitoring the change of ellipticity at 222 nm by CD spectroscopy (Appendix A). The reversible urea-induced denaturations monitored by far-UV CD of all the BRD mutants showed a sigmoidal dependence on denaturant concentration that could be fitted to a two-state model. The free energy of urea-induced unfolding, Δ*G*^H^_2_^O^, of the mutants and that of the wild type were obtained by globally fitting the whole dataset with a shared *m*-value of 1.93 kcal/mol/M [13] (Table 1). The mutants clustered into three classes according to their thermodynamic stability with respect to the wild type (ΔΔ*G*^H^_2_^O^). Mutants showing limited ΔΔ*G*^H^_2_^O^ included some mutants of the helix αZ (I356V, A366G), the loop ZA (A367G, A378G, A380G, L381A, and L383A), and the mutants A411G in the loop connecting αA and αB, as well as A437G in the helix αC. The mutants L350A, L357A, L360V, and L361A, located on the helix αZ, and the mutants V399A and V418A, located respectively, on the helices αA and αB, showed higher ΔΔ*G*^H^_2_^O^ values, ranging from 2 to 4 kcal/mol. All of these positions involved residues located in the hydrophobic core of the protein, suggesting that the destabilizing effects of the mutations on the thermodynamic stability may be referred to an alteration in the core of the protein. Finally, most of the mutants located in the C-terminal region of the protein (T398S, A415G, A416G, L420A, V435A, V436A, A439G, and V445A), as well as A369G and V376A, showed an intermediate decrease in thermodynamic stability with ΔΔ*G*^H^_2_^O^ values of 1 to 2 kcal/mol.

### 2.2. Folding Kinetics

As recalled above, we previously demonstrated that the two BET-BRDs, BRD2(2) and BRD4(1), follow a three-state mechanism, involving an on-pathway folding intermediate (I) in the sub-millisecond time regime [13]. In the same work, on the basis of a parameter derived from the kinetic data, i.e., the β-Tanford values [16,17], we estimated that, in the case of BRD2(2), the intermediate is characterized by limited compactness and poor native-like characteristics.

Therefore, in order to map in more detail the structural features of the transient species along the folding pathway of the BRD2(2) domain at a residue level and to identify the interactions stabilizing the intermediate and the folding transition state, we subjected all of the mutants to (un)folding kinetic experiments.

The complete (un)folding kinetics data set (chevron plot) obtained for each single mutant versus wild type BRD2(2) is reported in Figure 2 (representative kinetic folding and unfolding time courses are shown in Appendix A).

Following the same approach we have used in the characterization of the folding mechanism of BRD2(2) [13], the complete dataset was globally fitted to a three-state folding mechanism with shared values of *m* (Equation (4) in Materials and Methods). The calculated folding and unfolding parameters, together with the Φ values associated with the intermediate (I) and late transition state (TS2), are listed in Table 2. The robustness of our analysis was revealed by the good agreement between the Δ*G*_D-N_ values obtained from equilibrium denaturations (Table 1) and (un)folding kinetics (Table 2).

In accordance with the standard methodology [16], the calculated Φ values were divided into three groups (low: Φ < 0.3; intermediate: 0.3 < Φ < 0.7; high: Φ > 0.7) and mapped onto the native structure of BRD2(2). As shown in Figure 3a, the structural distribution of the Φ values indicated that the intermediate was characterized by a few native-like contacts identified by high Φ (I) values. Such a relatively small number of native-like interactions was in accordance with a non-compact partially folded state with scarce native-like characteristics, as hypothesized earlier on the basis of the low β-Tanford value [13]. The high Φ (I) values are located primarily in helix αB (A411, A415, A416), a region that represents the initial folding nucleus. As the number of native-like contacts was only marginally increased later in the process, it appears that the sequence of BRD2(2) was not optimized for efficient folding. Indeed, only two additional high Φ values were measured for the late transition state TS2 (A416 in helix αB and V399 in helix αA) (see Figure 3b). Interestingly, such a scenario, implying a rugged folding landscape, has been proposed earlier for another, unrelated, small four-helix bundle protein [18].

On the contrary, the N-terminal region of the domain (α-helix αZ and the ZA loop) appeared to be characterized by low Φ (I) values, suggesting that this region consolidates its structure only later in the folding process. Inspection of Table 2 shows that some Φ values (all involving residues in or interacting with the ZA loop) displayed unusually high values (i.e., Φ values > 1). Although, for two of them (A380G and L381A), the ΔΔ*G* was very low (<0.6 kcal/mol), thus precluding a reliable interpretation, the high Φ values of A439G (located in the C-terminal helix αC and establishing contacts with F372 in the ZA loop) observed for both the intermediate and TS2 suggest that this residue is involved in non-native interactions in both of these meta-stable states. Non-native interactions, as probed by unusual Φ values, have been found in other proteins, and it has been observed that they are often present in regions stabilizing folding intermediates [19] or in regions that are crucial for the function of the protein, such as the protein surfaces involved in recognition and binding [18,20]. In this context, it is interesting to note that the conformational plasticity of the ZA loop of BRDs, evidenced by molecular dynamics simulations, has been recently proposed to provide the necessary malleable interaction surface of the BRD domains to interact with their different target peptides [21].

In order to get an overall description of the structural and energetic properties of the intermediate and late transition state TS2, we resorted to analyzing the effects of the structural perturbations induced by mutagenesis by plotting the ΔΔ*G* of the intermediate (ΔΔ*G*_D-I_) and transition state TS2 (ΔΔ*G*_D-TS2_) versus those of the native states (ΔΔ*G*_D-N_) (Figure 4). This kind of analysis, known as Bronsted plot analysis [16], is commonly used to provide information on the folding landscape explored by proteins [14]. While a linear dependence is indicative of a pure nucleation mechanism, a more scattered Bronsted plot suggests the development of different nuclei, as predicted by a diffusion–collision folding mechanism [22]. Although the Bronsted plot for I and TS2 displayed an overall linear dependence, in the former case, dispersion of the data was more evident (R = 0.47 and 0.77, respectively) and the slope was lower (0.25 and 0.6, respectively), strengthening the hypothesis that the formation of the intermediate proceeds along a non-cooperative and rugged energy landscape, whereas a more cooperative process leads to the formation of the native-like transition state TS2. Inspection of Figure 4a shows that almost all of the positions with a higher value of ΔΔ*G*_D-I_ than the overall trend (i.e., residues 399, 411, 418, and 439) are clustered in the hydrophobic core of the protein domain, whereas the residues with the lower values of ΔΔ*G*_D-I_ (i.e., residues 350, 357, 367, and 378) are mainly located in the αZ helix and ZA loop. These results indicate that the stability of the intermediate does not rely on residues in the N-terminal part of the domain, but is mainly stabilized by a diffused nucleus involving a limited number of residues located in α-helices αA, αB, and αC in the C-terminal half of BRD2(2). On the contrary, the Bronsted plot for TS2 (Figure 4b) shows a better correlation, indicating that, as observed in other protein domains [23] and theoretically predicted [24], the late transition state TS2 is more native-like, representing a distorted version of the native state. Overall, these findings are in accordance with the distribution of the Φ values in the structure of BRD2(2) discussed above (Figure 3), and indicate that the intermediate is mainly stabilized by a small hydrophobic nucleus at the C-terminus of the domain and involves residues in the α-helices αA, αB, and αC.

## 3. Conclusions

Although the BRDs are protein domains that play crucial roles in many cellular processes, fundamental aspects, such as their folding mechanism, are still largely unexplored. The complete characterization of the folding of BRD2(2) by Φ value analysis provided in this work allowed us to obtain, for the first time in this protein class, structural information on both the intermediate I and transition state TS2. Moreover, by analyzing the contributions of native and non-native interactions at early and late stages of folding, we could depict a rugged folding landscape and hypothesize that the evolutionary pressure for maintaining the function of BRD2(2) may have decreased its folding efficiency. Future work will test this hypothesis by comparing the folding efficiency and binding properties of BRD2(2).

## 4. Materials and Methods

### 4.1. Site-Directed Mutagenesis

The constructs encoding the site-directed mutants of BRD2(2) were obtained using the gene encoding BRD2(2) wild type as a template to perform site-directed mutagenesis with the QuickChange Lightning Site-Directed Mutagenesis kit (Agilent Technologies, Santa Clara, CA) according to the manufacturer’s instructions. All mutations were confirmed by DNA sequencing analysis.

### 4.2. Protein Expression and Purification

The BRD2(2) wild type and all of the site-directed mutants were expressed in *E. coli* Rosetta cells. Bacterial cells were grown in an LB medium, containing 30 μg/mL of kanamycin at 37 °C until OD_600_ = 0.6, and then protein expression was induced with 0.5 mM of IPTG. After induction, cells were grown at 18 °C overnight and then collected by centrifugation.

To purify the protein, the bacterial pellet was resuspended and treated as described previously [25]. The purity of the protein was analyzed through SDS-PAGE, and the structural integrity of the purified proteins was checked by circular dichroism (CD) spectra in the far- and near-UV region. Protein concentration was determined spectrophotometrically using a molar absorptivity coefficient (ε 280) corresponding to 15,930 M^−1^cm^−1^ for wild type and the other mutants, based on a molecular mass of 13,351.3 Da, and calculated according to Gill and von Hippel [26].

### 4.3. Equilibrium Experiments

Equilibrium unfolding experiments were carried out at 20 °C in 20 mM of Tris HCl, pH = 7.5, 0.2 M of NaCl, and 200 μM of DTT. CD measurements were carried out with a JASCO J-720 spectropolarimeter using a 0.2 cm cuvette. BRD2(2) and all of the site-directed mutants, at a constant concentration of 80 µg/mL, were incubated at 20 °C at increasing urea concentration (0–9.0 M). When equilibrium was reached, far-UV CD spectra were recorded. The reversibility of the BRD2(2) wild type and mutant unfolding was checked as described previously [13]. All equilibrium unfolding experiments were performed in triplicate. Urea-induced equilibrium unfolding transitions monitored by far-UV CD ellipticity changes was analyzed by fitting the baseline and transition region data to a two-state linear extrapolation model [27] according to:Δ*G*_unfolding_ = Δ*G*^H^_2_^O^ + *m*[Urea] − RT ln (*K*_unfolding_)(1)
where Δ*G*_unfolding_ is the free energy change for unfolding for a given denaturant concentration, Δ*G*^H^_2_^O^ is the free energy change for unfolding in the absence of denaturant, *m* is a slope term which quantifies the change in Δ*G*_unfolding_ per unit concentration of denaturant, R is the gas constant, T is the temperature, and *K*_unfolding_ is the equilibrium constant for unfolding. The model expresses the signal as a function of denaturant concentration:yi = y_N_ + s_N_[X]i + (y_U_ + s_U_[X]i) ∗ exp[(−Δ*G*^H^_2_^O^ − *m*[X]i)/RT]/1 + exp[(−Δ*G*^H^_2_^O^ − *m*[X]i)/RT](2)
where yi is the observed signal, y_U_ and y_N_ are the baseline intercepts for unfolded and native protein, s_U_ and s_N_ are the baseline slopes for the unfolded and native protein, [X]i is the denaturant concentration after the ith addition, Δ*G*^H^_2_^O^ is the extrapolated free energy of unfolding in the absence of denaturant, and *m* is the slope in a Δ*G*_unfolding_ versus [X] plot. Data were globally fitted with the m-values shared between the datasets; all other parameters were not constrained.

The denaturant concentration at the midpoint of the transition, [Urea]_0.5_, according to Equation (2), is calculated as:[Urea]_0.5_ = Δ*G*^H^_2_^O^/*m*(3)

All unfolding transition data were fitted using GraphPad Prism 5. Data were normalized between 0 and 100%, where 0 corresponds to the molar ellipticity at 222 nm of the native protein, the smallest value (at 0 M urea), and 100 corresponds to the molar ellipticity at 222 nm of the unfolded protein, the largest value (at 9 M urea).

### 4.4. Kinetic Experiments

Unfolding and refolding kinetics experiments were performed using an SX-18 stopped-flow apparatus (Applied Photophysics, Leatherhead, UK). The protein samples were excited at 280 nm, and the fluorescence emission was measured using a 320 nm cutoff glass filter. The final concentration of the protein was typically 3 μM. At least five individual traces were acquired and then averaged for each experiment. All of the averages were satisfactorily fitted with a single exponential equation. Experiments were conducted in a buffer of 50 mM of Tris HCl, pH = 7.5, 0.2 M of NaCl, and 2 mM of DTT, as well as different concentrations of urea, ranging from 0.7 M to 8.1 M, at 20 °C.

The semilogarithmic plot (chevron plot) of each mutant was fitted on the basis of a three-state folding scheme with an on-pathway intermediate as previously reported [13,28] by using the following equation:Y_(X)_ = log((k_I-N_ * exp(-*m*_I-N_ * X/RT))/(1 + (1/K_D-I_ ∗ exp(*m*_D-I_ ∗ X/RT)) + k_N-I_ ∗ exp(*m*_N-I_ ∗ X/RT))(4)
where Y_(X)_ is the observed signal at the given denaturant concentration (X) and k_I-N_ and k_N-I_ represent the microscopic rate constants for the folding transition from the intermediate state (I) to the native state (N) and the unfolding from the native state (N) to the intermediate state (I) in the absence of denaturant, respectively; K_D-I_ is the equilibrium constant between the denatured (D) state and the I state, while *m*_I-N_, *m*_N-I_, and *m*_D-I_ are the denaturant dependence of the relative rate constants, R is universal gas constant, and T is the temperature expressed in Kelvin. During the global fitting procedure, the m-value was shared in the dataset to increase fitting accuracy. Data analysis was performed on GraphPad Prism software (GraphPad Software, Inc., San Diego, CA, USA).

The thermodynamic stability (Δ*G*_D-N_) for each mutant was evaluated from the equilibrium/kinetic constants summing the Δ*G*_D-I_ + Δ*G*_I-N_ stability as follow:Δ*G*_D-N_ = Δ*G*_D-I_ + Δ*G*_I-N_ = (-RT ln (1/K_D-I_)) + (-RT ln (k_I-N_/k_N-I_))(5)

The Φ values related to the intermediate state (Φ(I)) and the transition state 2 (Φ(TS2)) for each mutant were calculated as follows:Φ(TS2) = 1 − (ΔΔ*G*_TS2-N_/ΔΔ*G*_D-N_)(6)
Φ(I) = ΔΔ*G*_D-I_/ΔΔ*G*_D-N_(7)
where:ΔΔ*G*_D-N_ = Δ*G*^WT^_D-N_ − Δ*G*^mut^_D-N_(8)
ΔΔ*G*_TS2-N_ = Δ*G*^WT^_TS2-N_ − Δ*G*^mut^_TS2-N_ = RT ln (k^mut^_N-I_/k^WT^_N-I_) (9)
ΔΔ*G*_D-I_ = Δ*G*^WT^_D-I_ − Δ*G*^mut^_D-I_ = RT ln (k^mut^_D-I_/k^WT^_D-I_)(10)

## Figures and Tables

**Figure 1 ijms-22-05953-f001:**
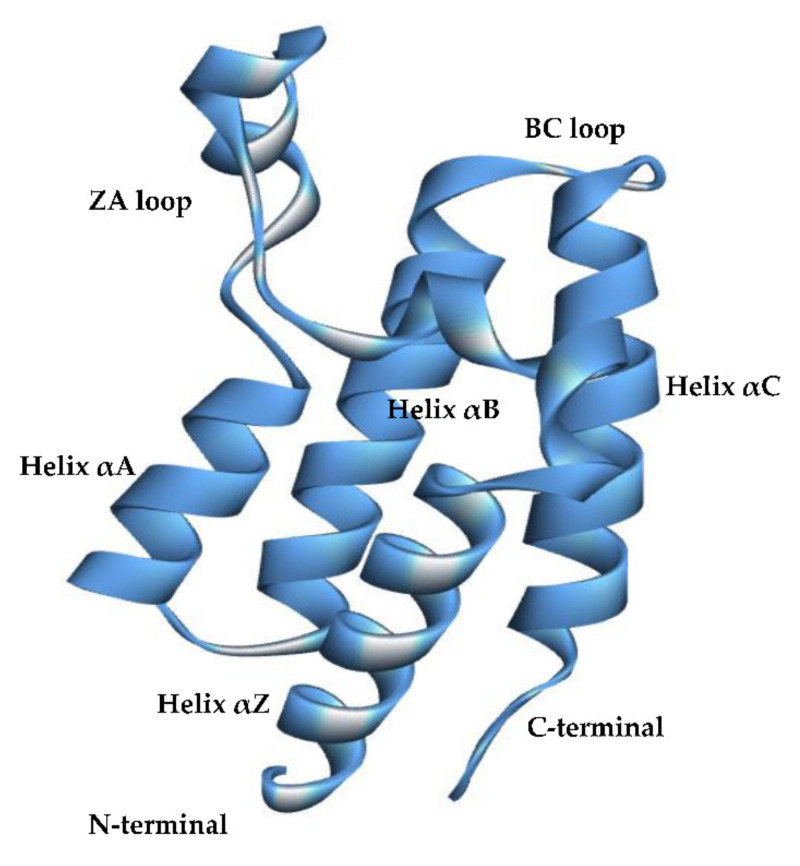
The structure of BRD2(2) (PDB: 3ONI). The structure of the bromodomains is a four-helix bundle formed by four conserved α-helices αZ, αA, αB, and αC connected by loop regions (ZA and BC loops) of variable length. The hydrophobic binding pocket is located at the end of the bundle (on top of the structure represented here) and surrounded by residues located on the loops ZA and BC.

**Figure 2 ijms-22-05953-f002:**
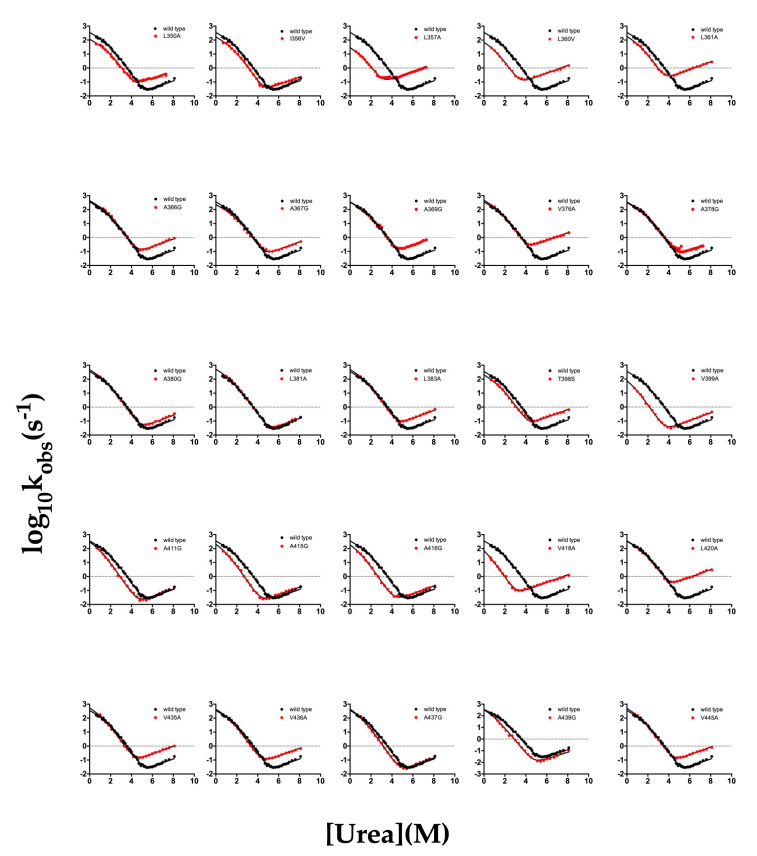
Chevron plots of BRD2(2) and its mutants. In the different panels the chevron plots of BRD2(2) wild type (in black) and its mutants (in red) are shown. All of the experiments were carried out in 50 mM of Tris HCl, pH = 7.5, 0.2 M of NaCl, and 2 mM of DTT. The data were globally fitted to a three-state folding mechanism, sharing the m-values.

**Figure 3 ijms-22-05953-f003:**
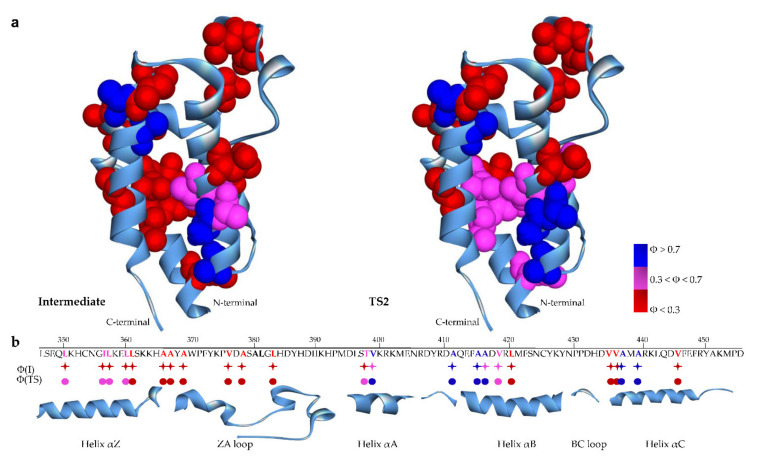
The structure and sequence of BRD2(2) (PDB: 3ONI). (**a**) Structural distribution of the measured Φ values on the structure of BRD2(2) in the intermediate (I) and in the transition state (TS2), respectively. The experimentally determined Φ values were divided into three categories and reported on the structure using the following code color: red, 0 < Φ < 0.30; magenta, 0.30 < Φ < 0.70; blue, 0.70 < Φ < 1.00. (**b**) Secondary structural elements are shown at the bottom of the amino acid sequence. Mutated residues are depicted in bold. The dots and asterisks under the sequence are colored according to the color code shown in (**a**).

**Figure 4 ijms-22-05953-f004:**
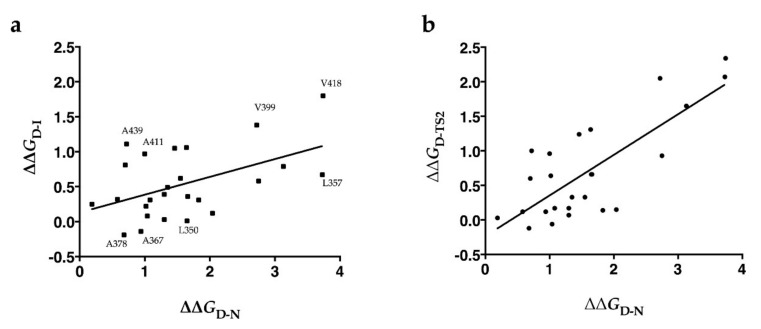
Bronsted plots for intermediate (**a**) (squares) and transition state two (**b**) (dots). In (**a**) the residues above/below the general trend for ∆∆*G*_D-I_ (see text) are also highlighted. The R values are 0.47 and 0.77, respectively, and the slope values are 0.25 and 0.6, respectively.

**Table 1 ijms-22-05953-t001:** Thermodynamic stability of BRD2(2) wild type and mutants.

Protein	Δ*G*^H^_2_^O^ (kcal/mol)	[Urea]_0.5_
wild type	8.94 ± 0.06	4.63
L350A	6.93 ± 0.06	3.59
I356V	8.24 ± 0.09	4.27
L357A	5.16 ± 0.09	2.67
L360V	6.37 ± 0.08	3.30
L361A	6.09 ± 0.05	3.15
A366G	8.31 ± 0.08	4.30
A367G	8.19 ± 0.06	4.24
A369G	7.88 ± 0.10	4.08
V376A	7.37 ± 0.08	3.82
A378G	8.49 ± 0.07	4.40
A380G	8.75 ± 0.10	4.53
L381A	9.23 ± 0.09	4.78
L383A	8.50 ± 0.08	4.40
T398S	7.57 ± 0.07	3.92
V399A	6.69 ± 0.07	3.47
A411G	8.44 ± 0.05	4.37
A415G	7.60 ± 0.11	3.94
A416G	7.77 ± 0.10	4.02
V418A	5.44 ± 0.04	2.82
L420A	6.97 ± 0.08	3.61
V435A	7.31 ± 0.08	3.79
V436A	7.81 ± 0.10	4.05
A437G	8.56 ± 0.11	4.43
A439G	7.54 ± 0.12	3.91
V445A	8.00 ± 0.08	4.14

Urea-induced unfolding equilibrium data were obtained at 20 °C in 20 mM of Tris HCl, pH = 7.5, containing 0.2 M of NaCl and 200 μM of DTT by measuring circular dichroism ellipticity at 222 nm [θ]_222_. Δ*G*^H^_2_^O^ values were obtained from Equation (2), [Urea]_0.5_ values were obtained from Equation (3). Data were globally fitted to a two-state model according to Equation (2), with the *m*-values shared between the datasets. Data are reported as the mean ± SE of the fit.

**Table 2 ijms-22-05953-t002:** Kinetic parameters of BRD2(2) and its mutants (see Materials and Methods) with Φ values calculation.

	k_I-N_ (s^−1^)	k_N-I_ (s^−1^)	K_D-I_	∆*G*_D-N_ (kcal/mol)	∆∆*G*_D-N_ (kcal/mol)	∆∆*G*_TS2-N_ (kcal/mol)	∆∆*G*_D-I_ (kcal/mol)	Φ (I)	Φ (TS2)
wild type	362 ± 32	7.44 × 10^−4^ ± 4.61 × 10^−5^	21 ± 2.7	9.37 ± 0.10					
L350A	118 ± 13	4.13 × 10^−3^ ± 2.59 × 10^−4^	21 ± 3.1	7.72 ± 0.11	1.65 ± 0.15	0.99 ± 0.05	0.01 ± 0.11	0.01 ± 0.11	0.40 ± 0.04
I356V	176± 21	1.43 × 10^−3^ ± 1.01 × 10^−4^	15 ± 2.5	8.36 ± 0.13	1.02 ± 0.16	0.38 ± 0.05	0.22 ± 0.12	0.22 ± 0.13	0.63 ± 0.13
L357A	32 ± 4.5	1.30 × 10^−2^ ± 7.01 × 10^−4^	6.7 ± 1.4	5.64 ± 0.14	3.73 ± 0.17	1.66 ± 0.05	0.67 ± 0.14	0.18 ± 0.04	0.55 ± 0.03
L360V	81 ± 14	9.53 × 10^−3^ ± 5.93 × 10^−4^	5.5 ± 1.4	6.24 ± 0.19	3.13 ± 0.21	1.48 ± 0.05	0.79 ± 0.17	0.25 ± 0.06	0.53 ± 0.04
L361A	200 ± 30	1.71 × 10^−2^ ± 1.09 × 10^−3^	7.8 ± 1.7	6.63 ± 0.16	2.75 ± 0.19	1.82 ± 0.05	0.58 ± 0.15	0.21 ± 0.06	0.34 ± 0.03
A366G	458 ± 54	4.92 × 10^−3^ ± 3.46 × 10^−4^	19 ± 3.1	8.33 ± 0.12	1.04 ± 0.16	1.10 ± 0.05	0.08 ± 0.12	0.08 ± 0.12	−0.05 ± 0.01
A367G	232 ± 26	3.07 × 10^−3^ ± 2.18 × 10^−4^	27 ± 3.5	8.43 ± 0.12	0.94 ± 0.16	0.82 ± 0.05	−0.14 ± 0.12	−0.15 ± 0.13	0.12 ± 0.02
A369G	340 ± 30	6.22 × 10^−3^ ± 4.02 × 10^−4^	20 ± 3.1	8.08 ± 0.12	1.30 ± 0.15	1.23 ± 0.05	0.03 ± 0.12	0.02 ± 0.08	0.05 ± 0.01
V376A	496 ± 42	1.38 × 10^−2^ ± 9.25 × 10^−4^	12 ± 2.3	7.54 ± 0.13	1.83 ± 0.16	1.69 ± 0.05	0.31 ± 0.13	0.17 ± 0.07	0.07 ± 0.01
A378G	319 ± 32	2.97 × 10^−3^ ± 1.95 × 10^−4^	30 ± 4.1	8.69 ± 0.11	0.68 ± 0.14	0.80 ± 0.05	−0.19 ± 0.11	−0.28 ± 0.17	−0.17 ± 0.04
A380G	502 ± 51	1.63 × 10^−3^ ± 1.19 × 10^−4^	12 ± 2.2	8.79 ± 0.13	0.58 ± 0.16	0.46 ± 0.06	0.32 ± 0.13	0.54 ^a^ ± 0.07	0.22 ^a^ ± 0.26
L381A	537 ± 56	9.83 × 10^−4^ ± 7.35 × 10^−5^	14 ± 2.4	9.19 ± 0.13	0.19 ± 0.16	0.16 ± 0.06	0.25 ± 0.12	1.36 ^a^ ± 0.12	0.13 ^a^ ± 1.28
L383A	461 ± 34	3.57 × 10^−3^ ± 2.23 × 10^−4^	12 ± 1.5	8.29 ± 0.10	1.08 ± 0.14	0.91 ± 0.05	0.31 ± 0.10	0.29 ± 0.05	0.16 ± 0.02
T398S	215 ± 27	4.18 × 10^−3^ ± 2.83 × 10^−4^	11 ± 2.2	7.71 ± 0.14	1.66 ± 0.17	1.00 ± 0.05	0.36 ± 0.13	0.21 ± 0.08	0.45 ± 0.05
V399A	114 ± 36	2.37 × 10^−3^ ± 1.51 × 10^−4^	1.9 ± 0.8	6.65 ± 0.11	2.72 ± 0.15	0.67 ± 0.05	1.38 ± 0.10	0.51 ± 0.05	0.75 ± 0.07
A411G	373 ± 30	7.97 × 10^−4^ ± 5.64 × 10^−5^	3.9 ± 0.7	8.37 ± 0.08	1.00 ± 0.12	0.04 * ± 0.05	0.97 ± 0.09	0.98 ± 0.15	0.96 ± 1.19
A415G	260 ± 24	1.08 × 10^−3^ ± 7.43 × 10^−5^	3.4 ± 0.7	7.91 ± 0.10	1.46 ± 0.14	0.22 ± 0.04	1.05 ± 0.11	0.72 ± 0.10	0.85 ± 0.16
A416G	237 ± 19	1.32 × 10^−3^ ± 8.96 × 10^−5^	3.4 ± 0.7	7.73 ± 0.10	1.64 ± 0.14	0.33 ± 0.05	1.06 ± 0.11	0.65 ± 0.09	0.80 ± 0.15
V418A	143 ± 13	8.38 × 10^−3^ ± 3.05 × 10^−4^	1.0 ± 0.4	5.63 ± 0.17	3.74 ± 0.19	1.40 ± 0.05	1.80 ± 0.17	0.48 ± 0.05	0.62 ± 0.04
L420A	348 ± 41	1.93 × 10^−2^ ± 1.28 × 10^−3^	17 ± 3.1	7.33 ± 0.13	2.04 ± 0.16	1.89 ± 0.05	0.12 ± 0.13	0.06 ± 0.06	0.07 ± 0.01
V435A	594 ± 56	6.07 × 10^−3^ ± 4.14 × 10^−4^	7.3 ± 1.6	7.82 ± 0.15	1.55 ± 0.18	1.22 ± 0.05	0.62 ± 0.14	0.40 ± 0.10	0.21 ± 0.03
V436A	479 ± 48	4.31 × 10^−3^ ± 2.66E × 10^−4^	9.2 ± 1.4	8.03 ± 0.11	1.35 ± 0.15	1.02 ± 0.05	0.49 ± 0.12	0.36 ± 0.09	0.24 ± 0.03
A437G	519 ± 35	8.83 × 10^−4^ ± 6.38 × 10^−5^	5.2 ± 1.2	8.67 ± 0.07	0.70 ± 0.12	0.10 ± 0.02	0.81 ± 0.10	1.16 ± 0.24	0.86 ± 0.17
A439G	436 ± 42	4.60 × 10^−4^ ± 3.33 × 10^−5^	3.2 ± 0.7	8.65 ± 0.11	0.72 ± 0.15	−0.28 ± 0.06	1.11 ± 0.11	1.54 ± 0.35	1.39 ± 0.39
V445A	527 ± 51	5.21 × 10^−3^ ± 3.23 × 10^−4^	11 ± 1.6	8.07 ± 0.11	1.30 ± 0.15	1.13 ± 0.05	0.39 ± 0.11	0.30 ± 0.09	0.13 ± 0.02

^a^ The mutant shows a ΔΔ*G*_D-N_ < 0.6 kcal/mol, preventing a reliable calculation of the Φ value. * The very low ∆∆*G*_TS2-N_ determines the high error on the Φ (TS2) value.

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
