# Peer review of "A Glimpse into the Structural Properties of the Intermediate and Transition State in the Folding of Bromodomain 2 Domain 2 by Φ Value Analysis"

_ijms, 2021, doi:10.3390/ijms22115953_

Round 1

Reviewer 1 Report

The work is quite schematic, however, the studies are done reliably and legibly presented. 

Minor note:
In the description of Figure 4 (Bronsted plots), it is useful to present statistical correlation parameters (especially parameter R). 

Reviewer 2 Report

The authors present an interesting study on folding/unfolding properties of Bromodomain 2 domain 2, a protein domain which seems to play a relevant role in eukaryotic gene expression. Furthermore, this domain represents a promising candidate to perform an extensive Φ-value analysis in order to elucidate relevant details of the folding process. The corresponding measurements have been conducted with care and the obtained results should have a good potential for meaningful conclusions. However, a few points need to be addressed with more attention before publication.

Major points:

CD-spectroscopy: Figure S1. It is not clear how the normalized ellipticity is defined. Typically, the CD signal (ellipticity) is a more or less strong negative signal in the native state. Upon unfolding the signal strength decreases. So why does Fig. S1 show an increasing signal upon unfolding? Maybe the authors should also show a representative CD-spectrum.

The authors characterized the equilibrium 2-state folding mechanism with CD- spectroscopy and the folding kinetics with tryptophan/tyrosine fluorescence spectroscopy. They should show that both techniques monitor the same folding process. It should be demonstrated at least that 3D integrity (fluorescence) and secondary structure integrity (CD) unfold at the same time /under the same conditions.

It is not fully clear how many states are really involved in the folding/unfolding transitions. On the one hand it is claimed the process is analyzed by the tree-state folding mechanism. On the other hand, the authors deal with four states: N, D. I, TS2? Most probable the TS2 cannot be resolved in time. The authors should explain this maybe in more detail.

The most important problem of this manuscript is related to the data interpretation, see page 8: The final interpretation relies on “dispersion of the data” and a “correlation of data points” as shown in the Bronsted plots. Both parameters are not defined and also not explained very well. It is not obvious for me that the data in Fig. 4b are better correlated than the data in Fig. 4a. Furthermore, the authors should also discuss the slope of the linear fitting lines (in Fig. 4), does this slope have a meaning?  

Minor points:

Page 9: Is the molecular mass of the investigated protein really 13351 kDa? This seems to be too high for a protein with 100 residues.

Fig. S2: Which experimental parameter is used for the values on the y-axis. How is this amplitude parameter derived from measured data (fluorescence emission intensity at which wavelength?). The authors should give for this shown example the results of the fitting procedures (model function, fitting conditions, i.e. fixed or free parameter, and obtained values including the error).  While the curve in Fig. S2a really looks like a mono-exponential, the curve in Fig. S2b looks more like a two exponential decay (the data on times above 0.05 show a linear decay behavior, which cannot be part of a mono-exponential decay)?!

Author Response

REVIEWER 2
Comments and Suggestions for Authors
The authors present an interesting study on folding/unfolding properties of Bromodomain 2 domain 2, a protein domain which seems to play a relevant role in eukaryotic gene expression. Furthermore, this domain represents a promising candidate to perform an extensive Φ-value analysis in order to elucidate relevant details of the folding process. The corresponding measurements have been conducted with care and the obtained results should have a good potential for meaningful conclusions. However, a few points need to be addressed with more attention before publication.
Major points:
1) CD-spectroscopy: Figure S1. It is not clear how the normalized ellipticity is defined. Typically, the CD signal (ellipticity) is a more or less strong negative signal in the native state. Upon unfolding the signal strength decreases. So why does Fig. S1 show an increasing signal upon unfolding? Maybe the authors should also show a representative CD-spectrum.
Response:
First of all we want to thank the Reviewer for his/her comments that help us to improve our paper.
For the equilibrium unfolding analysis we monitored the molar ellipticity changes at 222 nm between 0 M and 9 M urea. At increasing denaturant concentration, the protein unfolds as revealed by the progressive increase in the negative molar ellipticity at 222 nm ([θ]222]) that becomes progressively less negative. A plot of the molar ellipticity at 222nm as a function of denaturant concentration shows a sigmoid profile (Figure S1a). All the unfolding data were normalized between 0 and 100%, where 0 corresponds to the molar ellipticity at 222 nm of the native protein, the smallest value, (at 0 M urea), and 100 corresponds to the molar ellipticity at 222 nm of the unfolded protein, the largest value, (at 9 M urea). Normalization was calculated using GraphPad Prism 5 as follows:
z i = (x i – min(x)) / (max(x) – min(x)) * 100
where:
 z i : The i th normalized value in the dataset
 x i : The i th value in the dataset
 min(x): The smallest value in the dataset
 max(x): The largest value in the dataset
As suggested by the Reviewer we implemented the Figure S1 with representative CD spectra of BRD2(2) wild type. In particular, in Figure S1b are reported the representative far-UV CD spectra of native (0 M urea), at the midpoint of the unfolding transition (4.63 M urea, 50% unfolding) and unfolded (9 M urea) (100% unfolding) BRD2(2) wild type.
Furthermore, we added in the Matherials and Methods the following sentences:
All unfolding transition data were fitted using GraphPad Prism 5. Data were normalized between 0 and 100%, where 0 corresponds to the molar ellipticity at 222 nm of the native protein, the smallest value, (at 0 M urea) and 100 corresponds to the molar ellipticity at 222 nm of the unfolded protein, the largest value, (at 9 M urea). (p.10, line 278)
2) The authors characterized the equilibrium 2-state folding mechanism with CD- spectroscopy and the folding kinetics with tryptophan/tyrosine fluorescence spectroscopy. They should show that both techniques monitor the same folding process. It should be demonstrated at least that 3D integrity (fluorescence) and secondary structure integrity (CD) unfold at the same time /under the same conditions.
Response:
Concerning the suggestion of the Reviewer, we point out that the equilibrium CD spectroscopy experiments were carried out to measure their thermodynamic stabilities and not to investigate the folding mechanism. This information is indeed required to perform the following Φ-value analysis. As can be seen from Table 1 and 2, the equilibrium stability and the overall kinetic stability between the N and D states are perfectly comparable, strengthening the robustness of our experimental approach. Following the Referee’s suggestion, in the revised version at p.6 we added a column in Table 2 with the ΔGD-N values from kinetics and a sentence in the manuscript clarifying this point in the text (see in the Result Section 2.2, p.6 line 139).
The robustness of our analysis is revealed by the good agreement between ΔGD-N values obtained from equilibrium denaturations (Table 1) and (un)folding kinetics (Table 2).
3) It is not fully clear how many states are really involved in the folding/unfolding transitions. On the one hand it is claimed the process is analyzed by the tree-state folding mechanism. On the other hand, the authors deal with four states: N, D. I, TS2? Most probable the TS2 cannot be resolved in time. The authors should explain this maybe in more detail.
Response:
We wish to reassure the Referee about this point; we believe that the fact that this protein follow a three-state folding mechanism is correctly presented throughout the manuscript.
As detailed in the Introduction, the folding of the BRD2(2) can be described by a three-state mechanism as revealed previously (ref. Petrosino et al., 2017). The results obtained in our previous work showed that BRD2(2) can populate three states along the folding pathway: i) a fully unfolded state (D), ii) a transiently populated and partially folded intermediate state (I) iii) a native and fully folded state (N). Transition states (TS1 and TS2), being on top of the energy potential surface, are not considered thermodynamic states, which always correspond to global (or local) minima along the potential energy diagram. In particular, the transition state TS2 that we have characterized in the present work is the second energy barrier in the folding pathway of the protein, leading to the formation of N from the I state. The Φ-value analysis provided here by stopped-flow kinetic experiments, allowed us to measure the destabilization of I and TS2 caused by conservative single point mutations and to gain information about the structure of these states at near atomic resolution.
4) The most important problem of this manuscript is related to the data interpretation, see page 8: The final interpretation relies on “dispersion of the data” and a “correlation of data points” as shown in the Bronsted plots. Both parameters are not defined and also not explained very well. It is not obvious for me that the data in Fig. 4b are better correlated than the data in Fig. 4a. Furthermore, the authors should also discuss the slope of the linear fitting lines (in Fig. 4), does this slope have a meaning?
Response:
We apologize for not being clear enough about this topic.
The Bronsted plot is a powerful method to infer the overall structural features of metastable states in chemical reactions and relies on comparing the effects of structural perturbations on their free energies to those of the ground states.
In our work, the metastable states analyzed are the transiently populated intermediate state (I) and the experimentally inaccessible transition state 2 (TS2). In the Bronsted plot the free energy difference of these metastable states (ΔΔGD-I, and ΔΔGD-TS2) are reported as a function of ΔΔGD-N between wild type and each variant. The Bronsted plot analysis associated to folding studies is therefore often used to highlight two possible scenarios: on the one hand, a linear dependence between the ΔΔG of the given metastable state and ΔΔG of the overall folding stability indicates that all the mutations destabilize the metastable state to the same extent than the native state (slope of the linear regression fit of the data close to 1). This is indicative of a fully cooperative folding process (e.g. no folding intermediates). On the other hand, a poor dependence between the ΔΔG of the given metastable state and ΔΔG of the overall folding stability (slope of the fit close to 0) suggests a more dispersed and segregated folding process. Our results show that a higher slope can be observed for the late TS2 (slope ~0.6) prefiguring a more cooperative folding transition, while a slope of ~0.25 is obtained for the intermediate state (I), reflecting a more dispersed process.
As suggested by the Reviewer, we added in the legend to Figure 4 (p.9 line 217) the values of the slopes and the statistical correlation parameter (R) for both Bronsted plots and explained better in the text the meaning of these parameters.
Although the Bronsted plot for I and TS2 display an overall linear dependence, in the former case dispersion of the data is more evident (R = 0.47 and 0.77, respectively), and the slope is lower (0.25 and 0.6, respectively) strengthening the hypothesis that formation of the intermediate proceeds along a non-cooperative and rugged energy landscape, whereas a more cooperative process leads to the formation of the native-like transition state TS2. (p.8, line 194)
Minor points:
1) Page 9: Is the molecular mass of the investigated protein really 13351 kDa? This seems to be too high for a protein with 100 residues.
Response:
We agree with the Reviewer. We were wrong in the text, the correct molecular mass of BRD2(2) is 13351 Da. We apologize for our mistake and we substituted kDa with Da in the text accordingly. (p.9 line 249)
2) Fig. S2: Which experimental parameter is used for the values on the y-axis. How is this amplitude parameter derived from measured data (fluorescence emission intensity at which wavelength?). The authors should give for this shown example the results of the fitting procedures (model function, fitting conditions, i.e. fixed or free parameter, and obtained values including the error). While the curve in Fig. S2a really looks like a mono-exponential, the curve in Fig. S2b looks more like a two exponential decay (the data on times above 0.05 show a linear decay behavior, which cannot be part of a mono-exponential decay)?!
Response:
We thank again the Reviewer for helping us to improve our paper.
In our kinetic study we observed the change in fluorescence emission intensity as a function of time, exploiting the tryptophan of the BRD2(2) as a probe to follow the (un)folding process. The instrumentation used (stopped flow apparatus) allows us to select a cut-off filter (in this case a 320 nm glass filter) to follow the change in fluorescence intensity only at wavelengths above this value. The time courses (kinetic traces) were adequately fitted with a single exponential decay model (Y=a*e(-k*t) + c), from which
it is possible to obtain the amplitude (a: the difference between the final and the initial fluorescence signal), and the rate (k) of the trace.
As suggested by the Reviewer we added to the representative (un)folding traces (Fig. S2) the complete results of the fitting procedure and the residuals plot for both folding and unfolding time courses. Analysis of the residual plot clearly shows that the time course in Fig. S2a and b are adequately fitted by a single exponential decay.

Round 2

Reviewer 2 Report

can now be published